# Pregnant Women in Four Low-Middle Income Countries Have a High Prevalence of Inadequate Dietary Intakes That Are Improved by Dietary Diversity

**DOI:** 10.3390/nu11071560

**Published:** 2019-07-10

**Authors:** Rebecca L. Lander, K. Michael Hambidge, Jamie E. Westcott, Gabriela Tejeda, Tshilenge S. Diba, Shivanand C. Mastiholi, Umber S. Khan, Ana Garcés, Lester Figueroa, Antoinette Tshefu, Adrien Lokangaka, Shivaprasad S. Goudar, Manjunath S. Somannavar, Sumera Aziz Ali, Sarah Saleem, Elizabeth M. McClure, Nancy F. Krebs

**Affiliations:** 1Department of Pediatrics, Section of Nutrition, University of Colorado School of Medicine, Aurora, CO 80045, USA; 2Maternal and Infant Health Center, INCAP (Institute of Nutrition for Central America and Panama), Guatemala City 01011, Guatemala; 3Kinshasa School of Public Health, Kinshasa BP8502, Democratic Republic of the Congo; 4Women’s and Children’s Health Research Unit, KLE Academy of Higher Education and Research’s Jawaharlal Nehru Medical College, Belagavi, Karnataka 590010, India; 5Department of Community Health Sciences, Aga Khan University, Karachi 74800, Pakistan; 6RTI International, Durham, NC 27709, USA

**Keywords:** dietary assessment, nutrition, pregnant women, low middle income countries

## Abstract

Background: Up-to-date dietary data are required to understand the diverse nutritional challenges of pregnant women living in low-middle income countries (LMIC). To that end, dietary data were collected from 1st trimester pregnant women in rural areas of Guatemala, India, Pakistan, and Democratic Republic of the Congo (DRC) participating in a maternal lipid-based nutrient supplement (LNS) Randomized Controlled Trial to examine dietary diversity (DD), usual group energy and nutrient intakes, and prevalence of inadequate dietary intakes. Methods: Two 24-h dietary recalls were conducted in ~240 pregnant women/site (total *n* = 966) prior to 12-week gestation. Adequate DD was assessed, i.e., ≥5 major food groups consumed within the past 24 h. Median, Q1, Q3 intakes (without LNS) of energy, macronutrients, 12 micronutrients, and phytate were examined. The “at risk” prevalence of inadequate intakes were based on international guidelines for pregnant women. Results: Dietary patterns varied widely among sites, with adequate DD reported: 20% (Pakistan), 25% (DRC), 50% (Guatemala), and 70% (India). Significantly higher intakes of most key nutrients were observed in participants with adequate DD. More than 80% of women in all sites had inadequate intakes of folate, vitamin B12, and choline, and >80% of women in India and DRC also had inadequate intakes of calcium, thiamine, riboflavin, and vitamin B6. Conclusions: Our data highlight the likely need for micronutrient supplementation in pregnancy, specifically multi-micronutrient interventions, and support the value of increasing DD as part of sustainable long-term nutrition programs for women of reproductive age in these poor rural settings in LMIC.

## 1. Introduction

Poor maternal nutrition during pregnancy has multiple long-term adverse effects on mother and offspring health, including maternal anemia, postpartum complications, and increased neonatal morbidity and mortality [1]. Women of reproductive age (WRA) living in under-resourced environments in low-and middle-income countries (LMIC) are at particularly high risk of inadequate nutriture, especially of critical micronutrients [2], leaving them and their offspring ill-prepared for the increased nutritional demands of pregnancy and fetal growth. Embryogenesis and proper fetal development require sufficient dietary supplies of choline, folate, riboflavin, vitamin B6, and vitamin B12 to promote essential one-carbon metabolism for DNA methylation reactions [3,4]. Likewise, iron and zinc are necessary for development of the hippocampus and prefrontal cortex of the fetal brain, with even mild deficiency states associated with long-term neurocognitive risks [5]. Together with sufficient intakes of key micronutrients, a balanced and adequate supply of macronutrients, and thus energy, is also required for the physiological demands of the growing fetus [1]. Evidence strongly suggests that in the presence of maternal malnutrition, the offspring are at risk of decreased neonatal linear growth, i.e., stunting, with its associated heightened risk of long-term impaired cognition and delayed motor development [6].

In such environments with typically low-quality monotonous diets, multi-faceted nutritional approaches are required to improve birth outcomes. Recent lipid-based micronutrient supplementation (LNS) interventions during early gestation and throughout pregnancy have been implemented in several LMIC, with modest but positive effects observed, e.g., more adequate gestational weight gain and newborns with longer linear length [7,8]. Further, efforts are increasingly focused on improving the nutritional status of young women in these settings, who contribute to a large proportion of nulliparous births and have greater risks of preterm birth and low birth weight infants [9,10]. In this challenging milieu, more clarity of the current dietary context is critical to inform long-term programs.

The use of repeat 24-h dietary recalls has been successfully applied in LMIC [11] and can provide estimation of the prevalence of inadequate nutrient intakes at the population level, when conducted appropriately and in conjunction with a robustly compiled food nutrient composition database (FCDB). With these data, the diversity of the diet can also be assessed, e.g., the Minimum Dietary Diversity for WRA (MDD-W) has been recently developed as a proxy indicator to assess the micronutrient adequacy of women’s diets at a population level, particularly in LMIC [12]. Together, these measures can provide up-to-date information on the nutritional landscape and specific challenges faced in under-resourced areas.

The aims of the dietary assessment component of the Women First Preconception Maternal Nutrition Trial (WF), a large multi-national individually randomized preconception LNS randomized controlled trial (RCT) [8,13] conducted as part of the Global Network for Women’s and Children’s Health Research [14] in North and South Ubangi Province of Democratic Republic of the Congo (DRC), Western Highlands of Guatemala, Karnataka Province of India, and Sindh Province of Pakistan were to (1) examine the dietary diversity of the women’s diets, (2) estimate the usual group energy and nutrient intakes of 1st trimester pregnant women participating in the trial, and (3) determine the estimated prevalence of the study population “at risk” of inadequate nutrient intakes during 1st trimester.

## 2. Materials and Methods

### 2.1. Study Design

Repeat 24-h dietary recalls were conducted by a trained site nutritionist among pregnant women during their 1st trimester at each of the participating WF sites in DRC, Guatemala, India, and Pakistan from 2012 to 2017, as described earlier [13,15]. In brief, 3251 women in the WF study reached pregnancy and a randomly selected subset (*n* = approximately 240/site) were administered the dietary assessment (total dietary participants *n* = 966). Two 24-h dietary recalls were conducted 2–4 weeks apart once pregnancy was confirmed and prior to 12-weeks gestation. Of these women, half were randomly selected from Arm 1 (preconception supplementation) and half from Arm 2 (late 1st trimester supplementation), the latter group being assessed prior to commencing the intervention supplement. Dietary assessment training was provided for each of the site nutritionists by the lead study nutritionist (RL), with continuous support provided throughout the study.

### 2.2. Ethical Approval

Each participating research site received ethical approval for the conduct of this trial through their local institutional review board: DRC—Ecole de Sante Publique Comite d’Ethique 102B/14; Guatemala—Comite de Etica Universidad Francisco Marroquin 034-14; India—KLE Society’s JNMC Institutional Ethics Committee on Human Subjects Research MDC/IECHSR/2013-14/A25; Pakistan—Aga Khan University Ethical Review Committee 2753-CHS-ERC-13. The trial was registered with the US Office of Human Research Protection and with Federal Wide Assurances in place and Institutional Review Board approval at University of Colorado Anschutz Medical Campus. Written informed consent was obtained from all participants prior to participation.

### 2.3. Assessment of Food Intakes and Their Nutrient Adequacy

The two 24-h dietary recalls were conducted by the site nutritionist on non-consecutive days in the participant’s home, collecting data on all foods and beverages consumed within the past 24-h, as described earlier [15]. Nutritionists reported type of day (e.g., usual, feast, market, fasting); previous day’s health (e.g., well, nausea, vomiting); food consumption time (e.g., breakfast, lunch, dinner, or snack); location where food was consumed (e.g., in the home, outside); food/dish name; and amount consumed (grams). Dietary data were collected across all seasons at each site, with two seasons in DRC and Guatemala (i.e., harvest vs. dry) and three seasons in India and Pakistan (i.e., harvest, dry and rainy).

A unique FCDB was constructed at each site based on the food intake data collected from the dietary recalls [15]. In brief, the FCDB included moisture; macronutrients, protein, fat, fatty acids (i.e., total saturated, monounsaturated and polyunsaturated), and carbohydrate (CHO); and 12 micronutrients (calcium, iron, zinc, and vitamins A (RAE, retinol equivalent activity), thiamine, riboflavin, B6 (pyridoxal phosphate), folate (dietary folate equivalent), B12 (cobalamin), C (ascorbic acid), choline and betaine), as well as an anti-nutrient (phytate). Nutrient values were borrowed from regionally appropriate validated food composition tables [16,17,18,19,20], with >90% of foods/ingredients estimated to be an exact match to the local foods [21]. Particular care was taken regarding the consumption of fortified foods, especially in Guatemala where national fortification of wheat products with iron and B vitamins, as well as vitamin A-fortified sugar, is standard practice. Adjustments were made to all nutrient values, as needed, to account for differences in moisture content and nutrient retention and yield, using appropriate factors [22,23]. To calculate nutrient intakes from mixed dishes and beverages containing multiple ingredients, generic representative recipes were developed from the study population as described by Lander et al. [15], with more than 100 generic recipes compiled per site. Each site’s FCDB contained between 250 and 350 raw and cooked foods, beverages, and recipes in total.

Median, Q1, Q3 daily intakes of energy and selected nutrients were calculated, not including the LNS. The macronutrient distribution per site was compared to the acceptable macronutrient distribution range [24]. For protein and certain micronutrients, the prevalence of the risk of inadequate intakes was generated by comparison to the estimated average requirement (EAR) for the population by age and sex (i.e., pregnant females) [24]. Phytate:zinc molar ratios were calculated to provide an estimation of zinc absorption in relation to the phytate content of the diet, with low zinc bioavailability commonly reported in plant-based diets [11,25]. Dietary diversity scores were based on 10 primary food groups: starchy tubers and staples; pulses; nuts and seeds; milk/dairy; eggs; meat (including poultry and organs), fish, and insects; dark leafy greens; vitamin A–rich vegetables and fruit (≥60 µg RAE/100 g); other vegetables; and other fruit, as per the most recent MDD-W guidelines [12]. Information on consumption of additional food groups, including fats/oils; sweets; sugar sweetened beverages (SSB); and fast foods was also obtained. Low and adequate dietary diversity scores were calculated, based on consumption of <5 and ≥5 primary food groups, respectively [12].

### 2.4. Statistical Analyses

Dietary analyses were conducted by site including estimation of group usual energy and nutrient intake distributions and determination of the percentage of the population participating in the dietary assessment “at risk” of inadequate nutrient intakes compared to international guidelines by age and sex, using the EAR cut-point method [26]. Differences in median energy and nutrient intakes between relevant factors, e.g., study arm (Arms 1 vs. 2), age (16 to <19 years vs. 19–35 years), recall days (Recall 1 vs. Recall 2), reported health (well vs. vomiting), DD (low vs. adequate), and season (as defined by each site), were examined using the non-parametric Kruskal–Wallis test. Associations between study arm and DD, as well as age and DD, were examined using Fisher’s exact test. A *p*-value of <0.05 indicated statistical significance. Statistical analyses were performed using the STATA statistical software package 13 (Stata corporation, College Station, TX, USA). The Intake Monitoring, Assessment and Planning Program (IMAPP) Version 1.0 (Iowa State University 2010, Ames, Iowa, USA) was used to examine the “at risk” prevalence of inadequate intakes.

## 3. Results

From the overall WF study of 3251 women who entered the pregnancy stage, a subset of 966 1st trimester pregnant women were randomly selected and completed the dietary assessment: *n* = 218 (DRC), 230 (Guatemala), 245 (India), and 273 (Pakistan) (Table 1) with more than 1800 dietary recalls, including repeats, conducted in this study. The women ranged in mean age from 22.5 years (India) to 25.3 years (Guatemala) and at 12-week gestation, mean body mass index (BMI) (kg/m^2^) ranged from 19.9 (Pakistan) to 25.2 (Guatemala). The number of adolescent pregnant women (i.e., <19 years as defined by the World Health Organization) ranged from 5% (Guatemala) to 15% (Pakistan). At least 90% of women in all sites described the dietary assessment day as “usual,” with <2% of women in India and Pakistan observing fasting on the previous day, even during Ramadan. Few of the women (≤1%) in DRC and India reported feasting whereas ~10% of the Guatemalan and Pakistani women said feasting had occurred within the past 24 h. Specific questions were asked regarding pregnancy-related health symptoms with vomiting the previous day reported in <5% of women in DRC and Guatemala vs. 10–15% of women in Pakistan and India (Table 1).

No significant differences were found between nutrient intakes and study arm, age, or season. Reported health symptoms did not vary between recall day 1 and recall day 2. Significantly reduced intakes of most nutrients were observed in women complaining of vomiting in all sites, except Guatemala where no such associations were found.

### 3.1. Dietary Diversity (DD)

Only a fifth of Pakistani women consumed a diet with adequate DD (Figure 1). The main food groups contributing to energy intake consisted primarily of their staple of brown rice/wheat-flour chapattis (66%) and cow/buffalo milk tea and yogurt drinks (24%), with ≤3% contribution from other food groups (Figure 2). More than 70% of the Pakistani women who achieved adequate DD (i.e., *n* = 76 of 107 women) had consumed animal-source foods (ASF).

Likewise, only a quarter of the women in DRC reported adequate DD (Figure 1). Their main energy sources came from eating large amounts of fermented maize flour (stiff and soft varieties) and other staples (46%), e.g., cassava, and vitamin-A rich dark leafy greens (33%), and to a lesser extent non-vitamin A rich vegetables (7%), peanuts (7%), and small fish/insects (4%), e.g., caterpillars (Figure 2). Similar to the Pakistani women, 70% of the DRC women who attained adequate DD (i.e., *n* = 74 of 106 women) had eaten small fish, insects, caterpillars, and/or small amounts of flesh foods. Notably, red palm oil was a main ingredient in most DRC recipes.

Nearly 50% of Guatemalan women consumed a varied diet and had adequate DD (Figure 1). Frequent consumption of their staple foods, including maize flour tortillas, rice and bread (57%) contributed to their energy consumption, along with flesh foods (8%), pulses (5%), eggs (4%), non-vitamin A rich fruit (4%), and some dairy (2%), e.g., cheese (Figure 2). Like DRC and Pakistan, 70% of Guatemalan women who reported adequate DD (i.e., *n* = 155 of 221) had consumed ASF. However, unlike the other sites, approximately 20% of the energy intake in the Guatemalan diet came from sugar and refined processed foods, e.g., coffee with added sugar and approximately 10 varieties of local “atole” SSB, along with cakes, cookies, carbonated sodas, and potato chips.

In India, 70% of the women reported eating at least five food groups a day (Figure 1). Their staples of rice, wheat flour chapattis, and/or sorghum roti were the main energy contributors of the diet (46%), as well as buffalo milk tea and yogurt (18%), plus numerous types of pulses (12%), along with non-vitamin A rich vegetables (5%) and fruit (3%), as well as peanuts (3%) (Figure 2). Flesh foods were infrequently consumed, and fish was not part of the diet and thus, only 12% of women who achieved adequate DD (i.e., *n* = 39 of 329) had eaten ASF.

### 3.2. Median Energy and Nutrient Intakes

Median energy intakes by body weight were 37 kcal/kg/day (DRC), 34 kcal/kg/day (Guatemala), 33 kcal/kg/day (Pakistan), and 29 kcal/kg/day (India) (Table 2). Protein intakes for all sites were at the low end of the recommended intake (i.e., 0.88 g/kg/day), with higher median amounts of protein/kg/day consumed in Guatemala (1.04 g/kg/day) compared to Pakistan (0.82 g/kg/day), DRC (0.74 g/kg/day) and India (0.67 g/kg/day).

The median intake of the Guatemalan study population met the EAR for protein, as well as for calcium and vitamins A, riboflavin, and B6. For vitamin C, only the median intake of the group <19 years met the EAR (Table 3). Vitamin A-fortified sugar contributed substantially to vitamin A intakes, as did fortified wheat/maize flour and manufactured cereals to B vitamin consumption. The Guatemalan population with intakes at or above the 75th percentile for zinc and thiamine reached the EAR, and vitamin C for the group ≥19 years, largely due to substantial amounts of maize flour tortillas (zinc), fortified wheat bread (thiamine), and orange juice and fruit (vitamin C). Notably, approximately 2500 mg/day of phytate was consumed in the Guatemalan diet, reflected in the phytate:zinc molar ratio of 30.3.

In the Pakistani site participants, the median intake of vitamin B6 achieved the EAR, primarily from large consumption of brown rice flour chapattis. For protein, calcium, zinc, thiamine, riboflavin, and vitamin B12, the population of Pakistani women with intakes at or above the 75th percentile also reached the EAR, mainly due to large amounts of cow and buffalo milk (calcium, zinc, vitamin B12), chapattis (zinc, thiamine, riboflavin), and fish (zinc, vitamin B12) in their diet. Even though the median zinc intakes here were nearly two times greater than in India (8.8 vs. 4.7 mg/day, respectively), the phytate in the Pakistani diet was also nearly two times higher (2350 vs. 1090 mg/day, respectively) and thus, the phytate:zinc molar ratios for Pakistan and India are comparable, 23.3 vs. 22.9, respectively.

The Indian group failed to meet the EAR for any of the nutrients assessed, except for the portion of the population with protein intakes at or above the 75th percentile (0.88 g/kg/day). Notably, very low heme iron consumption was found in the India site participants (0.35 mg/day).

Similarly, only the DRC group with protein intakes at or above the 75th percentile met the EAR. However, the median intake of vitamin A exceeded the EAR, primarily because of the copious amounts of red palm oil used in most recipes. The median intake of vitamin C by the <19 years group met the EAR, mainly through consumption of large amounts of vitamin C-rich fruit, whereas only the study population ≥19 years with vitamin C intakes at or above the 75th percentile met the EAR. Lastly, DRC had the highest phytate:zinc molar ratio (59.4) of all the sites, with the lowest intakes of zinc (3.31 mg/day).

### 3.3. Estimated Prevalence of the Study Population “At Risk” of Inadequate Intakes

Significantly higher intakes of most key nutrients were observed in participants with adequate DD compared to inadequate DD for each site, including energy, protein, total fat, calcium, iron, zinc, vitamin A, folate, vitamin B12, and choline, except in Pakistan where fewer differences between the groups were noted (Table 4). Women who had consumed ASF had significantly higher intakes of iron (except DRC) and zinc (except Pakistan).

Overall, all sites except Guatemala had more than 80% of women at risk of inadequate protein. Further, more than 80% of women from all sites had a risk of inadequate intakes of folate, vitamin B12, and choline intakes (Figure 3, not all results shown). India and DRC also had >80% of women with inadequate intakes of calcium, thiamine, riboflavin, and vitamin B6. Assuming low bioavailability of iron (5%) in DRC, India and Pakistan due to their primarily plant-based diets, ≥98.5% of women in these three populations were at risk of inadequate iron intake. In contrast, 43% of the Guatemalan women were at risk of inadequate iron intake assuming intermediate bioavailability (10%) based on the larger amounts of ASF in their diet. It is of interest that DRC and Guatemala had the fewest women (<40%) with inadequate vitamin A consumption.

## 4. Discussion

The dietary data presented here from approximately 1000 1st trimester pregnant women from DRC, Guatemala, India, and Pakistan support a rationale for multi-micronutrient interventions in these populations, as well as for strategies to enhance dietary diversity, both to improve the adequacy of nutrient intakes. Few international nutrition studies have dietary data available from repeat 24-h recalls combined with robustly compiled site-specific FCDBs, and our results provide a reasonable estimate of quantitative nutrient intakes for these specific study participants in three regions of the world. These data are not only useful for ongoing international research in these four sites, but also informative for the strategic and appropriate upscaling of nutrition programs for vulnerable pregnant women living in similar under-resourced rural locations.

Our dietary results follow similar patterns previously reported for 2nd and 3rd trimester pregnant women in LMIC with higher energy intakes in the Latin American site (i.e., Guatemala) vs. South Asian sites (i.e., India and Pakistan) [27]. Indeed, the insufficient intakes of energy and nutrients observed in India and Pakistan are of potentially critical importance since poor maternal energy and micronutrient intakes are associated with impaired fetal development and may contribute to increased offspring visceral adiposity [28,29]. Our data are consistent with reports of low DD and/or sparse amounts of food consumed in these South Asian rural populations [30,31,32,33], and inadequate gestational weight gain compared to international guidelines was observed in these WF sites [8].

Notably, the prevalence of adequate DD is starkly different between India and Pakistan, 70% vs. 20%, respectively. While India had the highest DD of the four sites, the actual quantities of food consumed were very low and thus, very few of the Indian women met the EAR for most micronutrients. The meager amount of flesh foods in their diet, combined with low intakes, are reflected in the extremely low heme Fe intakes, as found elsewhere in southern India [34]. In contrast, despite the monotonous diet found in Pakistan, the major components of the diet, i.e., chapattis, dairy products and small amounts of flesh foods and fish, and modestly higher quantities of food consumed boosted intakes of calcium, iron, zinc, and most of the B vitamins, except folate.

The slightly higher rates of vomiting in these two sites are possibly reflected in the poor intakes reported here, but other major long-term factors may have also contributed, including poverty, low intra-household status, low maternal education, etc. Moreover, use of smokeless tobacco, e.g., “paan masala” (India) or “gutkha” (Pakistan), is common in these populations [35] which has been shown to contribute to reduced birth weight, preterm births, and degenerative placental changes in both Indian and Pakistani women [36,37]. Smokeless tobacco use was not quantitatively assessed here, yet its role as a potential appetite suppressant, may have affected intakes [38].

Of the four sites, the study population in DRC had the highest median energy intakes relative to body weight, and although their mean BMI is similar to Indian and Pakistani women, the African women are substantially taller [39]. Our data are very similar to intakes reported in a recent study of rural women in DRC [40], also based on two non-consecutive 24-h recalls, for energy, protein, fat, CHO, and most micronutrients, including calcium, iron, zinc, folate, and vitamin A. The nutritional vulnerability of the DRC diet is highlighted with low intakes of protein and most micronutrients, except possibly vitamin A. Finally, many rural African settings have reported a growing trend in WRA from underweight to overweight [41]. While our data do not suggest this transition, we are mindful of such risk.

Not entirely unexpectedly, our Guatemalan data highlights such a transition situation. The traditional foods of maize flour tortillas and beans are still a regular part of the diet, and the more diversified diet with larger serving sizes surely contributed to achieving the EAR for protein and a few key micronutrients, e.g., calcium and vitamin A. Yet the nutrition transition is clearly evidenced here, with a fifth of the energy consumed from processed energy-dense micronutrient-poor foods. This pattern has been seen in other pregnant Latin American women [27] and the large consumption of homemade traditional “atole” SSB consumed here is of particular concern. While vitamin A fortified table sugar is a principal source of vitamin A in the Guatemalan Highlands [42], liquid energy likely contributes to overconsumption via dysregulation of the link between sensory input and energy intake [43,44,45]. This may contribute to the observed overweight/obesity found in this population (mean BMI 25.2). An environment of excess energy has been associated with increased risk of offspring adiposity and long-term non-communicable diseases [46].

Adequate intakes of calcium, iron, zinc, folate, B12, and choline are required peri-conceptionally for the well-being of the embryo, not only to help ensure short-term survival but also to favor long-term health [1]. Apart from maternal prevention of eclampsia, sufficient calcium is also necessary to meet fetal bone requirements [47] and, except for Guatemala, none of the sites’ participants met the calcium EAR. Folate, vitamin B12, and choline are specifically involved in one-carbon metabolism, required for embryonic DNA methylation [4] and optimal epigenetic programming. Hence, it is of grave concern that >80% of women in all sites had inadequate 1st trimester intakes of these important nutrients. Further, the iron and zinc intakes in these women were far from robust, and both of these nutrients are essential for optimal fetal brain development [5]. Additionally, the phytate:zinc molar ratios are very high in these settings and, in contrast to later pregnancy and early lactation, the effect of these high ratios on zinc absorption is apparently not offset by up-regulation of zinc absorption early in gestation [48]. Thus, the poor dietary intakes/bioavailability of critical micronutrients support a rationale for preconception LNS or multiple micronutrient supplementation, as well as long-term nutrition programs specifically and strategically targeting adolescent and nulliparous women in these under-resourced environments [49].

The significant relationship observed here between higher intakes of most key nutrients and women with adequate DD is consistent with the validation work of the MDD-W indicator [50,51]. The data support use of DD as a proxy indicator for higher micronutrient adequacy in large surveys of WRA living in poor rural areas of LMIC [12]. Further, the noteworthy evidence of ASF contributing to higher intakes of iron and zinc reported here suggest that the “meat, poultry, and fish” food group of the MDD-W may play a unique role in achieving adequacy of these two crucial nutrients in the diet

Clearly, the challenges and opportunities faced by women in each study site are unique, and need fine-tuning on a population basis. Eggs, for example, were consumed in all sites but very infrequently (<10%), except in Guatemala. Population-specific promotion of increased intake of eggs could increase dietary diversity and potentially support improved maternal and offspring well-being [52]. In India, actively engaging local women in community education pertaining to amounts of food required in pregnancy combined with food-based approaches, e.g., development of iron-rich recipes from local green leafy vegetables, have been shown to be effective and are likely to be sustainable [53]. Food security in DRC is precarious yet this country has one of the most biodiverse environments in the world and the use of its wild edible plants has not been fully maximized as an affordable and sustainable source of dietary micronutrients [40]. Additionally, their regular consumption of insects, including caterpillars and termites, may provide an opportunity to increase intakes to further enhance protein intake. Various organizations are working with the Food and Agricultural Organization to identify and publish nutrient values for native foods [54], to encourage region-specific food-based approaches and integration into national dietary guidelines.

To this end, a challenge and limitation we faced in this study was determining the correct nutrient values for various wild species with the same local name. This limitation and others have been detailed in Lander et al. [15], including the challenge of accurately estimating portion sizes. DRC was the most difficult site to estimate amounts consumed because of the tradition of eating from the family pot, but our data are highly comparable to other recent dietary data reported from rural DRC [40]. In addition, we were unable to detect differences in seasonal intakes, as reported by others [55], perhaps because the two 24-h recalls here were conducted 2–4 weeks apart and may have straddled the seasonal cut-off established by the sites. For example, in DRC, if the first recall was in October and the second recall in November, they would have been coded harvest/winter vs. hunger/summer, respectively. We were unable to collect dietary data throughout the trial and thus cannot compare these data with 2nd and 3rd trimester intakes in these women. Lastly, we acknowledge the limitation of the 24-h dietary recall method and the potential for inaccurate reporting, where respondents may over- or under-report intakes in order to be viewed favorably by others. Nevertheless, we are confident our data provide reasonable estimations of usual group intakes for 1st trimester pregnant women due to the painstaking efforts to collect two 24-h non-consecutive recalls and the rigorous compilation of a unique FCDB for each site. Additionally, the continuous oversight and monitoring of all databases by the lead nutritionist (RL) assured high quality data.

## 5. Conclusions

In conclusion, this wealth of carefully collected dietary data, allowing for both quantitative and qualitative analyses in four distinctly different LMIC, provides a rich understanding of both the context-specific as well as the common nutrition challenges faced by pregnant women in these under-resourced settings. Our data indicate deficits in intakes of numerous micronutrients, and in some settings of energy and macronutrients, and thus support justification for supplementation prior to conception and in early pregnancy, specifically, multi-micronutrient interventions over iron-folate only supplementation. Targeted efforts to increase DD to enhance intakes of all nutrients are also warranted as part of strategic and sustainable long-term nutrition programs for WRA in poor rural settings in LMIC.

## Figures and Tables

**Figure 1 nutrients-11-01560-f001:**
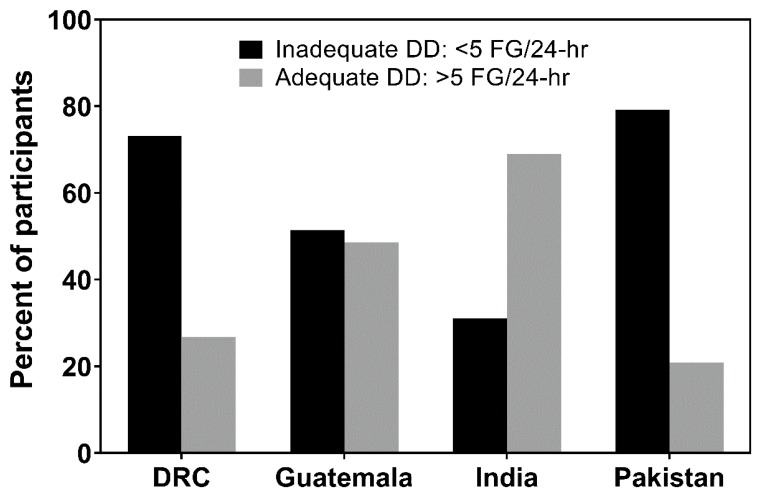
Inadequate and adequate dietary diversity (DD) ^1^ in pregnant women in four low- and middle-income countries by site. ^1^ Inadequate and Adequate DD, as defined by minimum dietary diversity for women of reproductive age [12], i.e., inadequate DD: consumption of <5 major food groups (FG) in the past 24 h; adequate DD: consumption of ≥5 major FG in the past 24 h. DRC, Democratic Republic of the Congo.

**Figure 2 nutrients-11-01560-f002:**
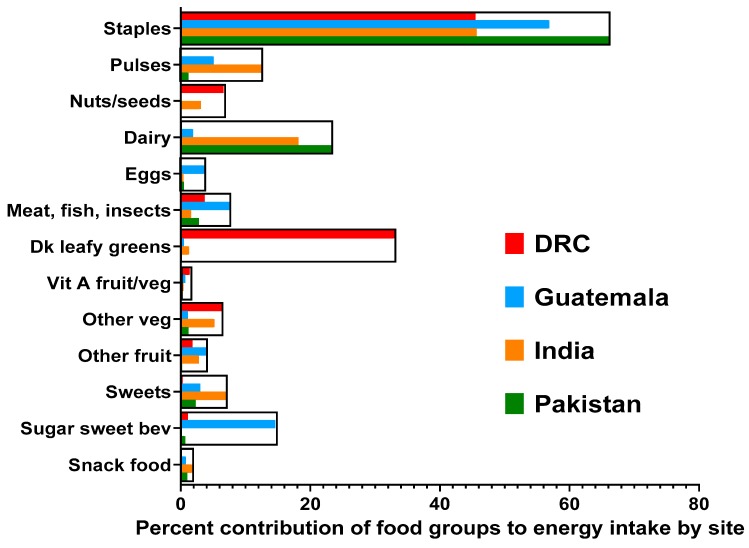
Contribution of food groups to total energy intake of pregnant women in four low- and middle-income countries by site. DRC, Democratic Republic of the Congo.

**Figure 3 nutrients-11-01560-f003:**
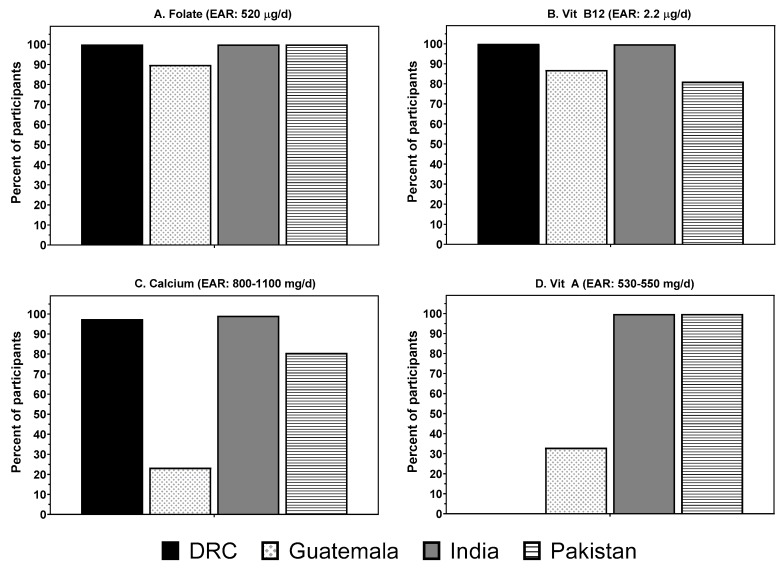
Estimated prevalence of the pregnant women in four low- and middle-income countries “at risk” of inadequate intakes of key micronutrients (**A**) Folate, (**B**) Vitamin B12, (**C**) Calcium, (**D**) Vitamin A. DRC, Democratic Republic of the Congo; Vit A, vitamin A; EAR, estimated average requirement; Vit B12, Vitamin B12.

**Table 1 nutrients-11-01560-t001:** Maternal demographics at 12-week gestation and dietary recalls by study arm in four low- and middle-income countries.

	DRC	Guatemala	India	Pakistan
*n*	218	230	245	273
Maternal demographics ^1^				
Age, years	24.2 ± 4.8	25.3 ± 4.1	22.5 ± 3.2	23.8 ± 4.3
<19 years (%)	13.7	5.3	6.3	15.0
≥19 years (%)	86.3	94.7	93.7	85.0
Weight, kg	51.0 ± 6.7	53.4 ± 9.2	46.2 ± 8.9	46.2 ± 7.3
Height, cm	156.1 ± 6.2	145.5 ± 4.9	151.4 ± 5.7	152.4 ± 6.3
BMI	20.8 ± 2.2	25.2 ± 4.0	20.3 ± 3.6	19.9 ± 3.0
Dietary Recalls (*n*)				
Arm 1—Day 1	99	107	117	136
Arm 1—Day 2	79	107	112	122
Arm 1 Total	178	214	229	258
Arm 2—Day 1	119	123	128	137
Arm 2—Day 2	98	118	120	120
Arm 2 Total	217	241	248	257
Total Recalls	395	455	477	515
Description of recall day, *n* (%)				
Usual	389 (98.5)	403 (88.6)	445 (93.3)	445 (86.4)
Fasting	0 (0)	0 (0)	11 (2.3)	3 (0.6)
Feasting	4 (1.0)	47 (10.3)	13 (2.7)	49 (9.5)
Market	2 (0.5)	5 (1.1)	8 (1.7)	18 (3.5)
Reported health of recall day, *n* (%)				
Vomiting	19 (4.8)	11 (2.4)	75 (15.7)	58 (11.2)
Nausea	20 (5.1)	206 (48.1)	99 (26.2)	26 (5.1)
Well	356 (90.1)	238 (49.5)	303 (58.1)	431 (83.7)

^1^ Values are presented as mean ± SD unless otherwise noted. DRC, Democratic Republic of the Congo; BMI, body mass index.

**Table 2 nutrients-11-01560-t002:** Median, Q1, Q3 energy and macronutrient intakes ^1^ in four low- and middle-income countries by site and comparison to the acceptable macronutrient distribution range (AMDR) ^2^.

	DRC	Guatemala	India	Pakistan
Energy, kcal/day	1821, 1258, 2449	1875, 1449, 2456	1281, 1047, 1600	1443, 1120, 1858
Energy, kcal/kg/day	37, 24, 51	34, 26, 47	29, 23, 36	33, 24, 42
Protein, g/day	36.6, 24.2, 51.4	54.6, 41.0, 74.1	30.5, 23.5, 39.6	36.9, 28.7, 47.9
Protein, g/kg/day	0.74, 0.48, 1.02	1.04, 0.73, 1.38	0.67, 0.49, 0.88	0.82, 0.62, 1.08
Total fat, g/day	83.4, 53.6, 122.0	37.1, 23.9, 55.3	51.7, 40.2, 65.4	33.7, 24.9, 44.8
CHO, g/day	215.5, 157.0, 292.9	342.6, 256.6, 441.6	179.1, 143.1, 220.7	244.2, 179.2, 306.7
Macronutrient distribution				
Protein (10–35%) ^2^	8%	11%	9%	10%
Fat (20–35%) ^2^	43%	17%	36%	21%
CHO (45–65%) ^2^	49%	71%	55%	68%

^1^ Values are median, Q1, Q3 unless otherwise noted. ^2^ AMDR Target Range: protein (10–35%); fat (20–35%); CHO (45–65%) [24]. AMDR, acceptable macronutrient distribution range; CHO, carbohydrate; DRC, Democratic Republic of the Congo.

**Table 3 nutrients-11-01560-t003:** Median, Q1, Q3 protein and micronutrient intakes ^1^ compared to the estimated average requirement (EAR) for pregnant women in four low- and middle-income countries.

Nutrient	EAR	DRC	Guatemala	India	Pakistan
Protein, g/kg/day	0.88 g/kg	0.74, 0.48, 1.02	1.04, 0.73, 1.38	0.67, 0.49, 0.88	0.82, 0.62, 1.08
Calcium, mg/day	1100 mg ^2^	498, 266, 720	1114, 606, 1320	395, 298, 603	524, 371, 658
	800 mg ^3^	433, 239, 640	956, 667, 1212	386, 251, 549	558, 359, 838
Total iron, mg/day	23 mg ^2^	9.18, 7.20, 13.81	14.15, 9.76, 18.89	8.20, 5.19, 10.65	10.12, 7.45,
	22 mg ^3^	8.74, 5.85, 13.28	13.31, 9.86, 17.71	7.62, 6.04, 9.70	9.58, 6.86, 15.69
Heme iron, mg/day ^4,5^	-	1.39 ± 2.52	2.77 ± 3.69	0.35 ± 1.21	1.80 ± 2.83
Zinc, mg/day	10.5 mg ^2^	3.42, 2.15, 5.95	9.24, 7.01, 12.40	4.81, 3.95, 6.83	8.63, 6.43, 11.07
	9.5 mg ^3^	3.30, 2.01, 5.19	7.91, 5.91, 10.55	4.71, 3.79, 6.03	8.88, 6.46, 11.43
Vit A (RAE), µg/day	530 µg ^2^	4876, 2956, 6353	1069, 848, 1374	227, 161, 350	152, 104, 202
	550 µg ^3^	3672, 2246, 5401	1050, 688, 1469	220, 155, 303	159, 110, 228
Thiamine, mg/day	1.2 mg	0.61, 0.43, 0.95	1.11, 0.79, 1.53	0.49, 0.39, 0.65	1.15, 0.79, 1.43
Riboflavin, mg/day	1.2 mg	0.68, 0.48, 0.98	1.25, 0.93, 1.63	0.75, 0.55, 0.98	1.00, 0.74, 1.39
Vit B6, mg/day	1.6 mg	0.82, 0.55, 1.31	1.65, 1.18, 2.29	0.97, 0.73, 1.24	1.79, 1.24, 2.48
Folate (DFE), µg/day	520 µg	184, 112, 276	348, 222, 489	122, 86, 177	90, 68, 123
Vit B12, µg/day	2.2 µg	0.01, 0.003, 0.44	0.85, 0.45, 1.62	0.73, 0.32, 1.24	1.19, 0.61, 2.03
Vit C, mg/day	66 mg ^2^	74, 39, 125	103, 56, 148	28, 22, 43	12, 8, 22
	70 mg ^3^	56, 28, 107	57, 25, 129	32, 19, 50	14, 9, 20
Choline, mg/day	450 mg ^4^	203, 142, 279	203, 110, 329	111, 84, 144	117, 85, 166
Betaine, mg/day	-	308, 155, 478	61, 23, 132	70, 48, 97	34, 11, 117
Phytate, mg/day	-	1807, 1057, 2941	2486, 1594, 3140	1090, 788, 1475	2350, 1339, 2553
Phytate:Zn molar ratio ^5^		59.38 ± 34.68	30.31 ± 10.24	23.25 ± 7.69	22.91 ± 4.70

^1^ Values are presented as median, Q1, Q3 unless otherwise noted. ^2^ EAR < 19 years [24]. ^3^ EAR ≥ 19–50 years [24]. ^4^ AI [24]. ^5^ Mean ± SD. AI, adequate intake; DFE, dietary folate equivalent; DRC, Democratic Republic of the Congo; EAR, estimated average requirement; RAE, retinol activity equivalents.

**Table 4 nutrients-11-01560-t004:** Median, Q1, Q3 nutrient intakes ^1,2^ for inadequate vs. adequate dietary diversity in pregnant women from four low- and middle-income countries by site.

	DRC		Guatemala		India		Pakistan	
Nutrient	Inadeq	Adeq	Inadeq	Adeq	Inadeq	Adeq	Inadeq	Adeq
*n* (%)	289	106	234	221	148	329	408	107
	(73.2%)	(26.8%)	(51.4%)	(48.6%)	(31.0%)	(69.0%)	(79.2%)	(20.8%)
Energy, kcal	1780	1990 *	1760	2010 *	1120	1320 *	1420	1550
	1140, 2420	1490, 2750	1300, 2290	1580, 2680	900, 1440	1090, 1650	110, 1850	1210, 1910
Protein, g	34.3	40.1 *	50.6	59.1 *	27.2	31.9 *	36.1	40.8 ***
	23.0, 49.5	30.7, 57.3	36.1, 67.3	45.1, 80.5	19.7, 35.2	24.6, 40.5	27.2, 47.3	31.6, 49.8
Fat, g	82.0	93.5 **	31.1	42.8 *	46.1	53.8 *	32.7	36.4 ***
	48.7, 114.5	65.5, 131.2	20.7, 49.9	31.1, 61.6	31.0, 57.7	42.4, 67.8	24.6, 44.3	27.9, 48.0
Calcium, mg	399	493 **	933	979	337	406 ***	540	594
	222, 618	282, 755	624, 1208	685, 1222	219, 526	276, 556	352, 808	401, 862
Iron, mg	8.42	9.89 **	12.75	14.21 **	6.96	8.08 *	9.36	10.23
	5.46, 12.52	6.70, 14.63	8.95, 16.63	10.39, 18.86	5.14, 8.68	6.35, 10.10	6.81, 15.52	7.42, 16.28
Zinc, mg	2.94	4.42 *	7.66	8.51 **	4.48	4.90 *	8.77	9.09
	1.88, 4.82	2.67, 6.20	5.65, 9.86	6.41, 11.32	3.36, 5.55	3.99, 6.35	6.34, 11.25	6.73, 11.73
Vit A, µg	3540	4130 **	931	1150 *	192	227 **	152	174 ***
	2050, 5280	2660, 6350	605, 1350	795, 1600	135, 268	163, 320	105, 217	116, 269
Folate, µg	173	227 *	303	377 **	97	134 **	88	97 ***
	107, 263	148, 344	210, 471	248, 516	68, 138	97, 191	64, 119	74, 133
Vit B12, µg	0.01	0.26 *	0.71	1.04 *	0.66	0.74	1.19	1.21
	0.002, 0.08	0.01, 1.04	0.22, 1.27	0.69, 1.86	0.27, 1.24	0.35, 1.24	0.60, 2.05	0.73, 1.93
Choline, mg	193	220 **	162	234 *	100	115 *	112	134 **
	126, 266	168, 307	89, 296	136, 353	69, 127	89, 150	82, 163	104, 180

^1^ Values are median, Q1, Q3 unless otherwise noted. ^2^ Non-parametric Kruskal–Wallis test comparing median intakes for inadequate vs. adequate dietary diversity; * *p* ≤ 0.0001; ** *p* ≤ 0.001; *** *p* ≤ 0.05. Adeq, Adequate; DRC, Democratic Republic of the Congo; Inadeq, Inadequate; Vit, vitamin.

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
