# Peer review of "Pregnant Women in Four Low-Middle Income Countries Have a High Prevalence of Inadequate Dietary Intakes That Are Improved by Dietary Diversity"

_nutrients, 2019, doi:10.3390/nu11071560_

Round 1
Reviewer 1 Report
In this manuscript, the authors performed a nutritional evaluation of the diet of pregnant women in four low income countries and evaluating the association with dietary diversity. They found a deficit risk for several micronutrientes, which was decreased when dietary diversity was higher. Therefore, promoting an increase in dietary diversity may be an strategy to reduce micronutrientes deficit during pregnancy in these countries.
I think the study was properly conceived and developed and there is a good discussion of the results. I have no specific suggestion.
Author Response
No comments to address
Reviewer 2 Report
The study describes the use of Dietary Diversity analysis and its relationship with nutrient intakes in four low- and middle-income countries. The study is interesting in the point of view that enhances the importance of diet diversity, and its promotion, as a tool to ensure adequate nutrient intake. However, some suggestions can be raised:
Abstract. Please describe the importance of the study in the background section.
Abstract. Line 31, describe what is IQR
Line 90. Include a flow diagram to describe how the volunteers were divided or selected. It is difficult to follow the significance of Arm 1 and Arm 2 with different days?
Line 101-104. Please include ethical approval number of each country site as well as the name of the Ethical Committee in each country site.
Lines 136. I believe that the food items included in the description/analysis of Diet Diversity in each are different between them since differences in eating patterns have been described between countries. It will be useful if the authors describe deeply, how the Diet Diversity was calculated in each country.
Line 174. Table 1. Please include other important demographic characteristics like the family number and the number of children in charge.
Line 239. Why do the authors decide to use the phytate: zinc molar ration instead of phytate:iron ratio?
Line 286. Table 4 is one of the principal results of the study. However, the authors do not deeply describe/discuss why the analysis of Dietary Diversity in Pakistan is not an appropriate tool to identify inadequate nutrient intake?
Line 333. Please include a reference in which it is evidenced that smokeless tobacco is an appetite suppressant
Conclusion of the study. Finally, the authors suggest that based on the data presented in this study it justified the need of nutrient supplementation programs, besides other efforts such as the enhancement of the diet diversification. It is well known that supplementation programs, although effective, are not sustainable through time. This point deserves a deep discussion in which it should include evidence of similar strategies with positive results and the sustainability in a short and long term
Author Response
The study describes the use of Dietary Diversity analysis and its relationship with nutrient intakes in four low- and middle-income countries. The study is interesting in the point of view that enhances the importance of diet diversity, and its promotion, as a tool to ensure adequate nutrient intake. However, some suggestions can be raised:
Abstract. Please describe the importance of the study in the background section.
- Please see additional comment in Lines 25-26: ‘Up-to-date dietary data are required to understand the diverse nutritional challenges of pregnant women living in low-middle income countries (LMIC). To that end, dietary data were collected…’
Abstract. Line 31, describe what is IQR
- It is the abbreviation for Interquartile range. The following has been added in Line 32: ‘Interquartile range’
Line 90. Include a flow diagram to describe how the volunteers were divided or selected. It is difficult to follow the significance of Arm 1 and Arm 2 with different days?
- The dietary assessment was conducted in a small subset of randomly selected women from Arm 1 and Arm 2 participating in the larger Women First RCT. We are unclear as to the question raised by the Reviewer. Thus, we would like to appeal that our wording is substantially clear in Lines 95-97: ‘…3251 women in the WF study reached pregnancy and a randomly selected subset (n=approximately 240/site) were administered the dietary assessment (total dietary participants n=966)….Of these women, half were randomly selected from Arm 1 (preconception supplementation) and half from Arm 2 (late 1st trimester supplementation), the latter group assessed prior to commencing the intervention supplement.’
Line 101-104. Please include ethical approval number of each country site as well as the name of the Ethical Committee in each country site.
- The following information has been added on Lines 102-105: ‘DRC – Ecole de Sante Publique Comite d’Ethique 102B/14; Guatemala – Comite de Etica Universidad Francisco Marroquin 034-14; India – KLE Society’s JNMC Institutional Ethics Committee on Human Subjects Research MDC/IECHSR/2013-14/A25; Pakistan – Aga Khan University Ethical Review Committee 2753-CHS-ERC-13.’
Lines 136. I believe that the food items included in the description/analysis of Diet Diversity in each are different between them since differences in eating patterns have been described between countries. It will be useful if the authors describe deeply, how the Diet Diversity was calculated in each country.
- Dietary Diversity was calculated in the same way in each country using the same food groups. Through using a standardized process, based on the guidelines from the Minimum Dietary Diversity for Women of Reproductive Age, we were able to detect different eating patterns between the countries.
Line 174. Table 1. Please include other important demographic characteristics like the family number and the number of children in charge.
- This information was not collected as part of the Women First study.
Line 239. Why do the authors decide to use the phytate: zinc molar ration instead of phytate:iron ratio?
- The influence of phytate on zinc absorption during stages of pregnancy and lactation has been documented in the literature. Hence, we emphasized the phytate:zinc molar ratio found in the diets of these first trimester pregnant women, as found in Lines 372-375: ‘Additionally, the phytate:zinc molar ratios are very high in these settings and, in contrast to later pregnancy and early lactation, the effect of these high ratios on zinc absorption is apparently not offset by up-regulation of zinc absorption early in gestation [47].’
Although phytate:iron ratio can be calculated, there are many other factors that influence iron absorption, including polyphenols in the foods, host iron status and host inflammatory status. These factors make the phytate:iron less useful, in contrast to the case for zinc, for which phytate is considered the single most important dietary factor (besides quantity of zinc intake) influencing zinc absorption.
Line 286. Table 4 is one of the principal results of the study. However, the authors do not deeply describe/discuss why the analysis of Dietary Diversity in Pakistan is not an appropriate tool to identify inadequate nutrient intake?
- While Pakistan had the lowest percentage of women with adequate dietary diversity (20%), we note in the manuscript that the quality of their monotonous diet enhanced dietary intakes in this population. Thus, fewer differences were noted in nutrient intakes between Pakistani women with adequate dietary diversity vs. inadequate dietary diversity as compared to the other sites. We discuss this finding in Lines 332-335: ‘In contrast, despite the monotonous diet found in Pakistan, the major components of the diet, i.e. chapattis, dairy products and small amounts of flesh foods and fish, and modestly higher quantities of food consumed boosted intakes of calcium, iron, zinc, and most of the B vitamins, except folate.’
Line 333. Please include a reference in which it is evidenced that smokeless tobacco is an appetite suppressant.
- We have changed the wording slightly and included a new reference in Lines 341-342: ‘Smokeless tobacco use was not quantitatively assessed here, yet its role as a potential appetite suppressant, may have affected intakes [38].
Conclusion of the study. Finally, the authors suggest that based on the data presented in this study it justified the need of nutrient supplementation programs, besides other efforts such as the enhancement of the diet diversification. It is well known that supplementation programs, although effective, are not sustainable through time. This point deserves a deep discussion in which it should include evidence of similar strategies with positive results and the sustainability in a short and long term
- Our intent in the final paragraph was to highlight the need for both types of interventions, including [short-term] multi-micronutrient supplementation and strategic, sustainable long-term nutrition programs, required in such under-resourced settings. We hesitate to include ‘short-term’ because it could mistakenly be interpreted to indicate a specific time-frame. Instead, we have included suggested parameters around the supplementation time, i.e. prior to conception and early pregnancy. Thus, we would like to appeal that the current wording is sufficient in Lines 422-427: ‘Our data indicate deficits in intakes of numerous micronutrients, and in some settings of energy and macronutrients, and thus support justification for supplementation prior to conception and in early pregnancy, specifically, multi-micronutrient interventions over iron-folate only supplementation. Targeted efforts to increase DD to enhance intakes of all nutrients are also warranted as part of strategic and sustainable long-term nutrition programs for WRA in poor rural settings in LMIC.’
Reviewer 3 Report
due to the differences of percentage in mothers<19 years between participant countries and also the differences in micronutrients recommendations for this population I would had appreciatted some discussion about the results in this age sub-group.
Author Response
Due to the differences of percentage in mothers<19 years between participant countries and also the differences in micronutrients recommendations for this population I would had appreciated some discussion about the results in this age sub-group.
- We agree with the Reviewer that it would have been of interest to specifically discuss the findings of mothers < 19 years of age. Unfortunately, the sample size of this group of women in our study was too small to draw accurate conclusions.